# Evaluating the capacity of large language models to interpret emotions in images

Hend Alrasheed[1,2*], Adwa Alghihab[2], Alex Pentland[2], Sharifa Alghowinem[2]

**1** Department of Information Technology, King Saud University, Riyadh, Saudi Arabia, **2** MIT Media Lab, Massachusetts Institute of Technology, Cambridge, Massachusetts, United States of America

* hrasheed@mit.edu

## Abstract

The integration of artificial intelligence, specifically large language models (LLMs), in emotional stimulus selection and validation offers a promising avenue for enhancing emotion comprehension frameworks. Traditional methods in this domain are often labor-intensive and susceptible to biases, highlighting the need for more efficient and scalable alternatives. This study evaluates the capability of GPT-4, in recognizing and rating emotions from visual stimuli, focusing on two primary emotional dimensions: valence (positive, neutral, or negative) and arousal (calm, neutral, or stimulated). By comparing the performance of GPT-4 against human evaluations using the well-established Geneva Affective PicturE Database (GAPED), we aim to assess the model's efficacy as a tool for automating the selection and validation of emotional elicitation stimuli. Our findings indicate that GPT-4 closely approximates human ratings under zero-shot learning conditions, although it encounters some difficulties in accurately classifying subtler emotional cues. These results underscore the potential of LLMs to streamline the emotional stimulus selection and validation process, thereby reducing the time and labor associated with traditional methods.

## Introduction

The integration of emotional stimuli within experimental settings is pivotal for probing emotional expressions in a replicable way, and therefore facilitating large scale assessments of emotional responses, not only in psychological research, but also for Affective Computing research. Selecting and validating emotional stimuli is critical for various applications ranging from basic emotion research and modeling, to clinical diagnostics and therapeutic interventions. According to [1], effective emotional stimuli should be contextually relevant, culturally sensitive, and capable of eliciting a broad spectrum of emotional responses. These stimuli should be standardized to ensure replicability and reliability across different studies.

Several frameworks have been developed to standardize emotion elicitation, each with its own methodologies and evaluation metrics [1–3]. Generally, to select and validate emotion stimuli, the researchers follow a multi-step process. First, the researchers identify a large number of potential stimuli that can elicit a range of emotions. The selected stimuli are then validated through a series of empirical studies, where thousand of participants are exposed to the stimuli, and their emotional responses are recorded and analyzed. Various methods such as

**Data availability statement:** The generated image descriptions 204 and their corresponding emotion ratings can be accessed at 205 https://github.com/halrashe/Emotional-LLMS.

**Funding:** The author(s) received no specific funding for this work.

**Competing interests:** The authors have declared that no competing interests exist.

self-report measures, physiological recordings, and behavioral observations are used to assess the emotional impact of the stimuli. Finally, the collected responses are statistically analyzed to narrow down the number of stimuli that were consistency and reliability effective in eliciting the intended emotions. Such methodologies emphasize the need for scalable and efficient solutions to overcome the limitations of traditional stimulus validation approaches, which are often time-intensive, labor-intensive and prone to subjective biases.

Images are one of the most commonly used forms of emotional stimuli due to their ease of presentation and ability to convey complex emotional content without linguistic barriers [4]. Numerous studies have employed visual stimuli to investigate various aspects of emotional processing. For example, one of the first published dataset is the International Affective Picture System (IAPS) [5], which has been extensively utilized in research to elicit emotional responses across different dimensions, including valence and arousal. Studies utilizing IAPS stimuli have consistently demonstrated its effectiveness in eliciting predictable and standardized emotional responses [6]. In addition, the Geneva Affective PicturE Database (GAPED) [4] offers a curated collection of images specifically designed to evoke a range of emotional responses. Research using GAPED has shown that these images reliably evoke strong emotional reactions across different populations, making it a valuable resource for emotion research [6]. The results from GAPED dataset [4] highlighted that despite cultural and linguistic variations, GAPED images can induce comparable emotional responses, underscoring the universality of certain emotional expressions. Datasets like GAPED and IAPS are based on the principle of emotion induction, making them valuable tools for emotion research. However, creating and maintaining such datasets on a large scale is a complex and resource-intensive process [6].

Large Language Models (LLMs), with their demonstrated proficiency in understanding emotions [7,8], offer a promising avenue for automating and refining visual emotional stimuli validation and analysis, ensuring rapid, objective, scalable, and consistent assessments. The predecessor version of OpenAI's LLM, GPT-3 [9], demonstrated a significant ability to discern emotions from textual data [10,11]. By leveraging the capabilities of GPT-4, this study aims to further explore the potential of LLMs in recognizing and rating emotions from images, thereby advancing emotion comprehension frameworks. We focus on two emotional dimensions: valence, which indicates the positivity or negativity of the perceived emotion, and arousal, which measures the level of excitement or calmness conveyed by the image. Our study benchmarks the performance of GPT-4 [12] against human evaluations using the Geneva Affective PicturE Database (GAPED) [4], offering insights into the model's processing of emotional content and its applicability in psychological contexts.

To achieve this, we conducted multiple experiments prompting GPT-4 to provide two types of ratings: numeric response ratings and Likert scale ratings, each under zero-shot and few-shot learning conditions. Additionally, a separate experiment was conducted where GPT-4 rated valence and arousal based on image textual descriptions, allowing for a comparative analysis of its performance across different input formats. While most efforts in the literature focus on evaluating the capabilities of LLMs in extracting emotions from facial images, our work focus to emotion recognition from general, non-facial images, such as objects, environments, animals, and abstract scenes. Despite their rich emotional content and widespread use in psychological, affective computing, and mental health research [4,6], the interpretation of emotions elicited by non-facial imagery remains relatively underexplored, particularly in the context of Large Language Models. Assessing emotional responses to such stimuli is crucial, as they provide opportunities to study affective processing in broader, more ecologically valid contexts where facial expressions may be absent or irrelevant. Moreover, non-facial images are

a foundational component of standardized emotional elicitation datasets such as GAPED and IAPS, highlighting their significance in emotion research.

Our results demonstrate that GPT-4 closely approximates human ratings under zero-shot learning conditions, achieving numeric response correlations of 0.87 for valence and 0.72 for arousal. For responses on a Likert scale, accuracies reached 0.77 for valence and 0.57 for arousal. Interestingly, incorporating examples directly into the prompts (few-shot learning) did not consistently enhance performance. This inconsistency may be due to the significant variation in human ratings across images within the same category. Additionally, the model encountered challenges in accurately classifying subtler emotional cues, suggesting areas for further refinement. Furthermore, we found that GPT-4' ability is comparable but slightly weaker in extrating emotions from the images textual description. Additionally, we observed that GPT-4's ability to extract emotions from textual descriptions of images is comparable but slightly less effective.

By demonstrating that GPT-4 can closely approximate human emotional ratings of visual stimuli, this work offers a scalable and efficient alternative to traditional emotion valida- tion methods, which are often labor-intensive and costly. Such automation can streamline experimental design in psychology and facilitate the creation of emotionally intelligent AI agents.

## Related work

With increasing interest in scalable and automated emotion elicitation approaches, studies have advanced from traditional methods to sophisticated AI-driven models, allowing for more efficient and robust emotion recognition across diverse datasets. This section reviews foundational and recent efforts in emotion elicitation, with a focus on image-based and text- based frameworks.

### Image-based emotion elicitation

Several studies have analyzed the sentiment and emotions that can be elicited from images [13–16]. Much of this research builds on earlier work in emotional semantic image retrieval [17,18], which aims to establish links between low-level image features and emotions to enable automatic image retrieval and categorization [6].

Early approaches to visual sentiment analysis predominantly relied on manual feature extraction, focusing on low-level visual attributes such as color, texture, and composition to infer emotional content [13,19]. In recent years, the field of emotion elicitation from visual images has increasingly adopted AI tools to enhance the accuracy and efficiency of emotion recognition. Convolutional neural networks, for instance, have the ability to automatically learn and extract complex, high-level features directly from raw image data [20–23]. Addi- tionally, reinforcement learning has been adopted to fine-tune pre-trained models on specific datasets to improve performance in domain-specific emotion analysis [24–26].

Recent advancements have enabled Large Language Models (LLMs) to handle multi- modal inputs, such as text, images, and video, making them highly general-purpose tools. In [27], the authors compare the performances of deep learning models with LLMs for image- based emotion recognition using a facial expression image dataset. While deep learning mod- els, such as convolutional neural networks and other specialized architectures, were trained specifically for this task, LLMs were evaluated on the same dataset without any specialized fine-tuning. Among the LLMs, the best-performing model achieved an accuracy of 55.8%, outperforming some deep learning models but falling short of the top specialized models.

Despite not surpassing the best deep learning models, LLMs demonstrated competitive performance, particularly with smaller datasets, positioning them as a viable option for specific tasks without requiring extensive additional training.

The study in [8] explores the ability of ChatGPT-4 [12] and Google Bard [28] to interpret emotional cues from both visual and textual data. For visual emotion recognition, the authors employed the Reading the Mind in the Eyes Test [29], which includes 36 images of the eye region of human faces. For textual emotion analysis, they used the Levels of Emotional Awareness Scale [30], consisting of 20 open-ended questions designed to evoke emotional responses. The findings show that ChatGPT-4 excelled in recognizing emotional cues from visual data, achieving scores comparable to human benchmarks and demonstrating no biases related to gender or emotion type. In contrast, Google Bard's visual recognition performance was near random, with significantly lower scores. Both models, however, performed exceptionally well in interpreting emotions from text, exceeding human averages.

The study in [31] provides a quantitative evaluation of GPT-4V's performance across 21 benchmark datasets for various tasks related to emotion recognition, including visual sentiment analysis, micro-expression recognition, facial emotion recognition, dynamic facial emotion recognition, and multimodal emotion recognition. The findings demonstrate that GPT-4V has strong visual understanding capabilities, effectively integrating multimodal cues and temporal information, both essential for emotion recognition. However, the model struggles with micro-expression recognition (involuntary facial movements revealing people's hidden feelings), a task requiring specialized knowledge. While GPT-4V outperforms random guessing, it falls short compared to supervised systems.

In [32], the authors explore the potential of Large Language Models (LLMs) for enhancing emotion recognition using facial data by leveraging multiple learning conditions. They investigate three key approaches: fine-tuning on emotion datasets, as well as zero-shot and few-shot learning for scenarios with limited or no labeled data. The authors utilize In-Context Learning and Chain-of-Thought reasoning to enhance the interpretability and accuracy of LLMs in emotion recognition, enabling the models to provide more informed predictions. Extensive experiments demonstrate that LLMs, even without parameter updates, deliver competitive results in emotion recognition tasks, often outperforming traditional methods.

While most efforts in the literature focus on evaluating the capabilities of LLMs in extracting emotions from facial images, our work shifts the focus to emotion recognition from general, non-facial images. Accordingly, we use the GAPED image dataset, a well-established resource for emotion elicitation. Several studies have employed this dataset for a range of emotional research tasks. For example, Moyal et al. [33] used GAPED images to elicit discrete emotions such as fear, disgust, sadness, and happiness. Balsamo et al. [34] used valence and arousal ratings to evaluate the effectiveness of GAPED images in evoking emotional responses. Moreover, the author in [35] used GAPED images to explore how different combinations of valence and arousal influence cognitive processing, particularly in emotionally ambiguous contexts. A recent study [36] used the GAPED dataset to assess valence and arousal ratings within a Malaysian population using a 9-point Likert scale, with the goal of identifying culturally specific patterns in emotional responses.

## Text-based emotion elicitation

Numerous studies have explored the extraction of emotions from text, with early work focusing on sentiment analysis and emotion classification using lexical resources and traditional machine learning techniques [37–39]. These approaches often rely on manually designed features, such as word frequency, polarity lexicons, and syntactic structures, to infer emotional

content. A prominent example is the use of associations of words with specific emotions like anger, joy, or sadness, allowing for rule-based emotion detection from textual data [39].

More recently, deep learning models have revolutionized the field by automatically learning complex patterns in text. Recurrent neural networks and their variants have been widely adopted to capture the sequential nature of text and detect emotions based on contextual word embeddings [40,41]. Transformer-based models, particularly BERT and its variations, have significantly advanced emotion elicitation by leveraging attention mechanisms to understand the relationships between words over long sequences. These models, trained using supervised learning techniques, have achieved state-of-the-art results in emotion classification and sentiment analysis tasks [42,43]. They are typically fine-tuned on specific labeled emotion datasets to enhance their emotion recognition performance in a supervised learning setting [44].

In addition to supervised approaches, unsupervised and semi-supervised techniques have been proposed to handle scenarios where labeled data is limited. These methods use large pre-trained models combined with transfer learning and self-supervised objectives, allowing them to perform well even in domains with little emotion-labeled text [45,46].

LLMs have demonstrated commendable performance in sentiment analysis tasks [47–51]. Their emergence has significantly advanced emotion elicitation from text, as they possess a deep understanding of natural language semantics and context.

In a recent study, [47] evaluated the alignment between LLMs and human emotions and values using a novel psychometric text-based assessment called Situational Evaluation of Complex Emotional Understanding, specifically tailored for evaluating Emotional Intelligence across both human participants and LLMs. Their findings revealed that most LLMs achieved above-average scores on this assessment, with GPT-4 notably surpassing 89% of human participants, underscoring its potential in understanding and reflecting human emotional and value-based nuances.

The study presented in [50] aims to evaluate the effectiveness of large language models (LLMs), specifically GPT-4, in estimating concreteness, valence, and arousal for multi-word expressions, which are essential for understanding emotional and cognitive responses to language. Using the GPT-4o model, researchers compared its predictions with human ratings on concreteness from a prior study and found high correlation scores, indicating that the model accurately reflected human judgments. The study further extended these assessments to multi-word expressions, where GPT-4o continued to produce reliable estimates, underscoring the model's potential in psycholinguistic research.

In [51], the potential of GPT-4 for automating emotion annotation in a zero-shot setting is examined. The study used four publicly available emotion recognition datasets: first-hand emotional reports labeled with basic emotions, a Twitter dataset with various emotion classes, a Reddit-based dataset featuring a broad range of emotion categories, and a multi-genre English dataset annotated in the valence-arousal-dominance space. Results from human evaluation experiments consistently indicated a preference for GPT-4's annotations over those by human annotators across multiple datasets and evaluators.

## Materials and methods

The primary goal of this study is to evaluate the accuracy of Large Language Models (LLMs), particularly GPT-4, in perceiving emotions in images and their textual descriptions, specifically across the dimensions of valence and arousal. We utilize the Geneva Affective PicturE Database (GAPED), a dataset of images that have been rated for emotional content by human participants.

To assess the performance of GPT-4, we conducted two groups of experiments. In the first group, we prompted GPT-4 to provide valence and arousal ratings for each image in the GAPED dataset. These ratings were then compared directly to the human-generated ratings to evaluate the models' accuracy in interpreting emotions from visual stimuli. In the second group, we investigated the ability of the GPT-4 to infer emotions from textual descriptions of the same images. For this, we first prompted GPT-4 to generate detailed textual descriptions of the images. These descriptions were subsequently used as inputs to the model, which provided emotion ratings based on the text alone. This allowed us to explore how well the models can capture emotional content from text as compared to direct visual input. The generated image descriptions and their corresponding emotion ratings can be accessed at https://github.com/halrashe/Emotional-LLMS.

## Image dataset

We utilize the Geneva Affective PicturE Database (GAPED) [4], which contains 730 images, classified into three primary categories: positive, neutral, and negative. The positive category includes images of human and animal infants, as well as nature scenes. Neutral images typically depict inanimate objects. The negative category is further divided into four subclasses: animal mistreatment (featuring scenes of animal cruelty), human concerns (depicting violations of human rights), snakes, and spiders. Table 1 summarizes the number of images in each category along with their average valence and arousal ratings as evaluated by human participants.

Each image in the GAPED dataset was rated by human evaluators across two emotional dimensions: valence and arousal. These evaluations were provided by sixty participants from a second-year psychology class, with an average age of 24 years. The participants, though predominantly native French speakers, represented diverse cultural backgrounds.

The ratings, expressed as decimal values from 0 to 100, represent the emotional responses elicited by the images. Valence scores reflect the emotional tone, where 0 corresponds to very negative emotions and 100 to very positive ones, with 50 representing a neutral response. Arousal scores, on the other hand, measure emotional intensity, where 0 indicates calmness and 100 represents high stimulation, with 50 denoting neutrality.

In the GAPED dataset, positive images generally exhibit high valence scores (70 and above) paired with low arousal scores (below 22). Negative images tend to have lower valence scores (under 50) and moderately higher arousal scores (ranging from 53 to 61). Neutral images typically register valence scores around 55 and arousal scores approximately 25. The image categories have some overlap, i.e., certain images in the Negative category were rated

**Table 1. Number of images in each category of the dataset along with their average valence and arousal ratings provided by human participants. The table also illustrates the distribution of human ratings for valence and arousal using the Likert scale.**

| Image category | Number of images | Average valence | Average arousal | Valence | | | Arousal | | |
|---|---|---|---|---|---|---|---|---|---|
| | | | | Positive | Neutral | Negative | Calm | Neutral | Stimulated |
| All images | 730 | 44.5 | 47.7 | 17% | 35% | 48% | 16% | 25% | 59% |
| Positive | 121 | 89.6 | 21.6 | 100% | 0% | 0% | 62% | 35% | 3% |
| Neutral | 89 | 55.8 | 24.9 | 0% | 100% | 0% | 44% | 56% | 0% |
| Animal mistreatment | 124 | 21.3 | 60.6 | 0% | 11% | 89% | 0% | 11% | 89% |
| Human Concerns | 105 | 28.0 | 58.7 | 0% | 30% | 70% | 0% | 23% | 77% |
| Snakes | 133 | 41.5 | 53.6 | 0% | 59% | 41% | 0% | 26% | 74% |
| Spiders | 158 | 35.1 | 58.2 | 0% | 28% | 72% | 0% | 15% | 85% |

Neutral (for valence or arousal, or both) by human raters. Despite this overlap in rating ranges, the positive, neutral, and negative categories display clear emotional distinctions.

We implemented two key updates to the image dataset. First, we added a textual description to each image, generated by GPT-4, by prompting the model with "What's in this image?" to describe the content of the image. Second, we assigned valence and arousal ratings on a 3-point Likert scale to each image. Valence was categorized as Negative, Neutral, or Positive, while arousal was classified as Calm, Neutral, or Stimulated. These ratings were derived by mapping the original continuous valence and arousal values to the Likert scale based on the ranges established by human raters in [4]. The valence scale was divided as follows: Negative [0,40), Neutral [40,71), and Positive [71,100]. For arousal, Calm [0,23), Neutral [23,45), and Stimulated [45,100].

It is important to note that this new rating system may lead to changes in the original classification of some images. For instance, an image initially categorized as Negative may now be classified as Neutral based on its updated valence score derived from human ratings. Table 1 illustrates the distribution of human ratings for valence and arousal using the Likert scale.

## Valence and arousal ratings for images

We used GPT-4 Turbo [12] for all prompts assessing emotional perception from images. The model was administered two types of rating questions: one requiring numeric response ratings and the other using a Likert scale format.

- Numeric response rating: GPT is prompted to generate numeric ratings for valence and arousal on a scale from 0 to 100, emulating human ratings from GAPED. The following prompt was used:

  *You are participating in a visual emotion perception study focused on understanding how different visual scenes affect emotions. As you view the uploaded image, consider its emotional impact.*

  *Valence: Rate the emotional tone of the image on a scale from 0 (very negative) to 100 (very positive), with 50 indicating neutrality. Arousal: Assess the emotional intensity of the image on a scale from 0 (very calm) to 100 (very stimulated), with 50 representing a neutral state.*

  *Provide only your estimated ratings for Valence and Arousal for the image, formatted as: "Valence: [value], Arousal: [value]".*

- Likert scale rating: GPT is prompted to provide ratings based on a 3-point scale (Negative, Neutral, Positive for valence; Calm, Neutral, Stimulated for arousal). A similar prompt to the one used for numeric ratings was employed, but it included the specific rating options.

  *You are participating in a visual emotion perception study focused on understanding how different visual scenes affect emotions. As you view the uploaded image, consider its emotional impact.*

  *Valence: Rate the emotional tone of the image on a scale Negative, Neutral, Positive. Arousal: Assess the emotional intensity of the image on a scale Calm, Neutral, Stimulated.*

  *Provide only your estimated ratings for Valence and Arousal for the image, formatted as: "Valence: [value], Arousal: [value]".*

For each rating type, two learning conditions were used to evaluate the model's performance: *zero-shot prompting*, where the model generates responses based solely on its pretrained knowledge without any specific examples, and *few-shot prompting*, where a small set

of example images is provided to guide the model's responses. In the few-shot setting, the prompt included randomly selected images from each category: 3 from Positive, 3 from Neutral, and a total of 8 from Negative (2 from each sub-category), along with their corresponding human ratings. See S1 Fig for prompt templates used in few-shot prompting.

We conducted four assessments: (1) numeric response ratings using zero-shot prompting, (2) numeric response ratings using few-shot prompting, (3) Likert scale ratings using zero-shot prompting, and (4) Likert scale ratings using few-shot prompting. To ensure consistency in numeric ratings, results are based on the average of 10 responses for numeric response ratings. For Likert scale ratings, the most frequent response from 9 prompts was used to determine the final rating, with 9 prompts chosen to avoid the possibility of ties.

## Valence and arousal ratings for image descriptions

Similar to the image rating tasks, GPT-4 Turbo was used for all text-based assessments, where the model was given two types of rating questions: numeric response ratings and Likert scale ratings. In these assessments, each text input was a previously GPT-generated description of an image, with the model relying solely on the description rather than the image itself.

- Numeric response rating: GPT is prompted to generate numeric ratings for valence and arousal on a scale from 0 to 100. The following prompt was used:

  *You are participating in a visual emotion perception study focused on understanding how different visual scenes affect emotions. As you read the following image description, consider its emotional impact.*

  *Valence: Rate the emotional tone of the image description on a scale from 0 (very negative) to 100 (very positive), with 50 indicating neutrality. Arousal: Assess the emotional intensity of the image description on a scale from 0 (very calm) to 100 (very stimulated), with 50 representing a neutral state.*

  *Provide only your estimated ratings for Valence and Arousal for the image description, formatted as: "Valence: [value], Arousal: [value]".*

- Likert scale rating: GPT is prompted to provide ratings based on a 3-point scale (Negative, Neutral, Positive for valence; Calm, Neutral, Stimulated for arousal). A similar prompt format was used but adapted to the rating options.

We employed both zero-shot and few-shot prompting, with few-shot prompts including a set of text-based examples similar to those used in the image assessment.

## Results

### Valence and arousal ratings for images

**Numeric response rating.** GPT-4's performance in numeric response rating tasks for valence and arousal compared to human raters shows strong correlations. In the zero-shot setting, the Pearson correlation is $r = 0.87$ ($p<0.001$) for valence and $r = 0.72$ ($p<0.001$) for arousal. In the few-shot setting, the correlations are $r = 0.86$ ($p<0.001$) for valence and $r = 0.80$ ($p<0.001$) for arousal. Tables 2 and 3 provide a detailed breakdown of these results, including the mean, standard deviation, and range of ratings given by human raters in the GAPED dataset, alongside GPT-4's ratings under both zero-shot and few-shot learning conditions.

Overall, our findings demonstrate that GPT-4 closely approximates human ratings for both valence and arousal in both zero-shot and few-shot settings. As shown in Table 2, the comparison between GPT-4's zero-shot mean and the human mean indicates that GPT-4 performs remarkably well without any prior task-specific training. Moreover, the consistently low

**Table 2. Numeric response ratings for valence across images under zero-shot and few-shot learning conditions. SD represents standard deviation; MAE represents mean absolute error.**

| Image category | Humans | | | GPT-4 zero-shot | | | | GPT-4 few-shot | | | |
|---|---|---|---|---|---|---|---|---|---|---|---|
| | Mean | SD | Range | Mean | SD | Range | MAE | Mean | SD | Range | MAE |
| All images | 44.5 | 25.2 | 0.4-98.7 | 41.7 | 19.0 | 3.5-85.0 | 10.5 | 45.6 | 20.2 | 7.0-94.2 | 10.2 |
| Positive | 89.6 | 6.2 | 71.9-98.7 | 75.8 | 5.7 | 61.5-85.0 | 13.9 | 83.0 | 7.1 | 64.8-94.2 | 8.5 |
| Neutral | 55.8 | 6.1 | 41.0-68.9 | 50.1 | 5.2 | 33.2-65.2 | 6.7 | 52.5 | 5.1 | 42.6-67.4 | 5.5 |
| Animal Mistreatment | 21.3 | 12.4 | 0.4-49.5 | 26.9 | 12.4 | 8.5-69.0 | 9.5 | 30.3 | 11.4 | 8.3-71.7 | 11.7 |
| Human Concerns | 28.0 | 17.5 | 0.7-61.4 | 35.4 | 17.1 | 3.5-73.4 | 11.0 | 40.1 | 17.3 | 7.0-85.4 | 14.5 |
| Snakes | 41.5 | 11.2 | 16.7-63.7 | 36.2 | 5.9 | 15.7-58.0 | 11.4 | 39.2 | 6.7 | 19.1-68.6 | 10.7 |
| Spiders | 35.1 | 11.4 | 9.5-57.0 | 31.2 | 4.8 | 20.0-60.6 | 9.7 | 34.2 | 5.7 | 17.9-60.0 | 9.6 |

**Table 3. Numeric response ratings for arousal across images under zero-shot and few-shot learning conditions. SD represents standard deviation; MAE represents mean absolute error.**

| Image category | Humans | | | GPT-4 zero-shot | | | | GPT-4 few-shot | | | |
|---|---|---|---|---|---|---|---|---|---|---|---|
| | Mean | SD | Range | Mean | SD | Range | MAE | Mean | SD | Range | MAE |
| All images | 47.7 | 19.5 | 5.9-92.4 | 49.7 | 17.0 | 10.0-86.5 | 11.1 | 47.7 | 18.1 | 10.0-81.8 | 9.5 |
| Positive | 21.6 | 10.7 | 5.9-66.0 | 34.4 | 8.1 | 20.5-76.5 | 13.3 | 26.0 | 7.2 | 15.4-65.6 | 7.8 |
| Neutral | 24.9 | 7.8 | 10.2-43.8 | 21.9 | 7.8 | 10.0-40.3 | 7.3 | 20.2 | 5.6 | 10.0-36.3 | 6.8 |
| Animal Mistreatment | 60.6 | 12.0 | 28.4-89.0 | 54.7 | 13.0 | 23.5-81.3 | 10.9 | 56.3 | 11.1 | 21.8-81.8 | 9.4 |
| Human Concerns | 58.7 | 15.0 | 32.3-92.4 | 48.8 | 13.9 | 21.5-86.5 | 13.3 | 49.2 | 14.2 | 20.1-81.0 | 12.2 |
| Snakes | 53.6 | 10.7 | 30.6-72.9 | 62.1 | 5.3 | 37.0-76.1 | 11.7 | 59.7 | 6.3 | 28.9-69.7 | 11.1 |
| Spiders | 58.2 | 10.3 | 37.4-78.4 | 63.3 | 5.0 | 32.0-73.0 | 9.6 | 61.8 | 4.9 | 32.0-74.4 | 9.3 |

Mean Absolute Error (MAE) across all image categories suggests that the model's predictions align closely with human assessments, with only minor deviations. For instance, the valence MAE for all image ratings indicates that GPT-4's predictions deviate from human ratings by an average of 10.5 points. Note that the maximum possible deviation between GPT-4 and the human rating for any image is 100, as both valence and arousal were rated on a 0–100 scale. In our results, MAE values ranged from approximately 5 to 15, indicating that the average error across all image categories was relatively small, representing only 5–15% of the maximum possible error.

This performance improves with the inclusion of examples in the few-shot setting for most image categories. For instance, the Mean Absolute Error (MAE) for valence in Positive images decreases from 13.9 in the zero-shot setting to 8.5 in the few-shot setting, as shown in Table 2. Similarly, for Neutral images, the MAE drops from 6.7 to 5.5. However, in some categories of Negative images, the inclusion of examples did not improve performance and, in certain cases, even resulted in slightly worse predictions compared to the zero-shot setting.

Similar trends are observed for arousal, as detailed in Table 3, where GPT-4's performance improves with the inclusion of examples in the few-shot setting. For all images, the Mean Absolute Error (MAE) decreases from 11.1 in the zero-shot setting to 9.5 in the few-shot setting. Notably, for Positive images, the MAE drops significantly from 13.3 to 7.8. Comparable enhancements are observed across the other categories.

Fig 1 shows the distribution of differences in valence and arousal ratings between GPT-4 and human raters across all images and individual image categories, using kernel density estimates. Positive differences indicate that GPT-4 rated the emotional impact (either valence or arousal) higher than human raters, while negative differences suggest lower ratings by GPT-4. Each peak within the plots represents the most commonly observed difference. For example, the All Images panel in Fig 1 illustrates the close alignment between human and GPT-4 ratings for both valence and arousal, with peaks near zero indicating strong agreement. In the

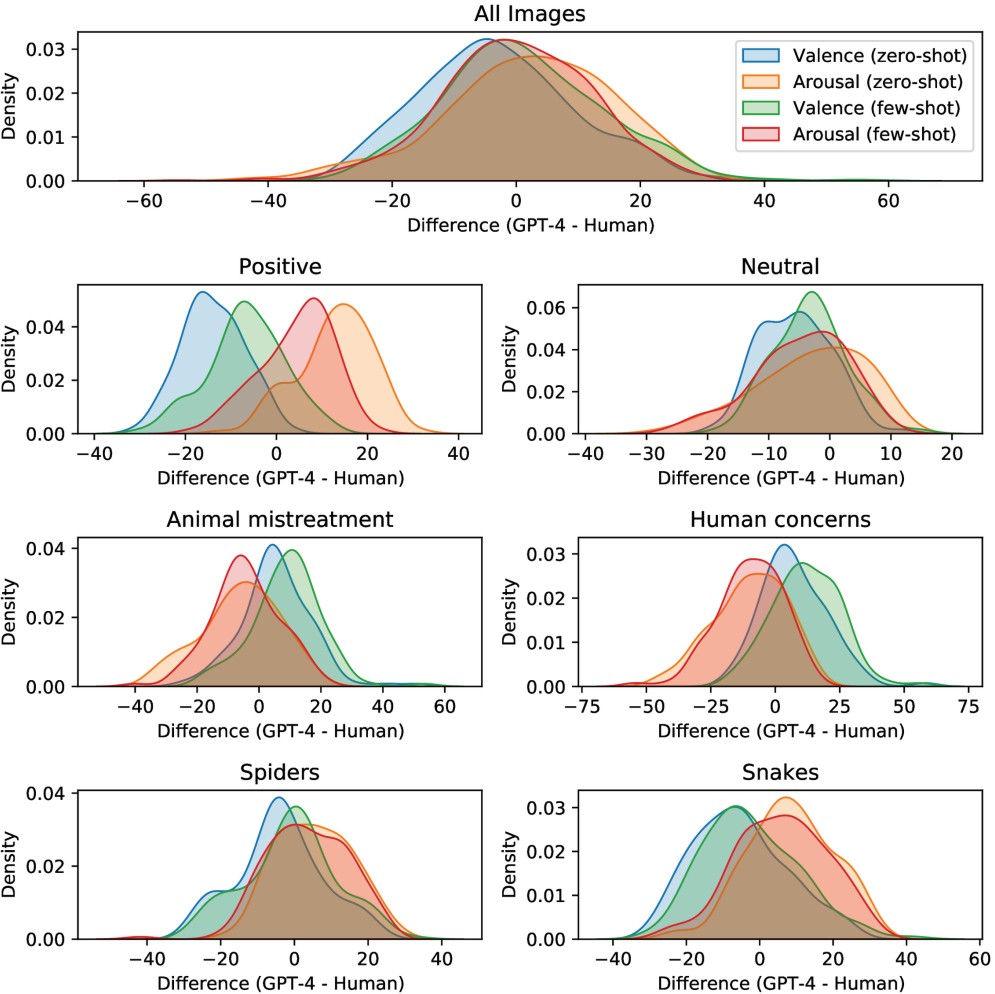

**Fig 1. Distributions of rating differences across image categories.** Distributions of valence and arousal rating differences between human participants and GPT-4 predictions across image categories.

Positive image category, the few-shot model's ratings show greater alignment with human ratings, as the few-shot peaks for valence and arousal are closer to zero compared to those in the zero-shot setting. In the Animal mistreatment panel of Fig 1, a prominent peak around +5 indicates that humans generally assign a stronger negative valence to these images compared to GPT-4, while a peak near -5 reveals that GPT-4 tends to provide slightly lower arousal ratings for these images. Similar patterns appear in the Human Concerns panel. Conversely, in the Snakes and Spiders panel, GPT-4 shows a tendency to assign stronger negative valence and heightened arousal relative to human ratings. Fig 2 shows examples with close, moderately close, and divergent ratings between human raters and GPT-4, within each image category.

**Likert scale rating.** Fig 3 shows the agreement between human and GPT-4 valence and arousal ratings of all images on the 3-point Likert scale under zero-shot and few-shot conditions. Each matrix highlights the frequency of predicted ratings compared to actual human ratings, illustrating areas of alignment and discrepancy in emotional impact perception across various prompts.

| Close agreement | Moderate agreement | Low agreement |
|---|---|---|

**Positive**

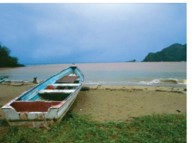

Human Valence: 84.6
Human Arousal: 26.2

GPT-4 Valence: 82.5
GPT-4 Arousal: 35.6

The image shows a cute kitten with fluffy white and grey fur sitting inside a wooden bucket. Next to the bucket there is a pot filled with purple flowers possibly lobelia. The kitten has striking blue eyes and is looking upwards giving it an adorable curious expression. The background includes a wooden surface which adds a rustic charm to the scene.

Human Valence: 79.2
Human Arousal: 17.3

GPT-4 Valence: 62.1
GPT-4 Arousal: 30

The image shows a scenic landscape featuring a beach. In the foreground there is an old weathered boat with peeling paint resting on the sandy shore. The background reveals a calm sea meeting the overcast sky. Additionally lush greenery is visible along the edges of the beach and there are mountains faintly visible in the distance partially obscured by mist or low clouds. This tranquil somewhat moody scene seems to be isolated with no visible human activity.

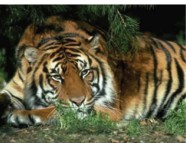

Human Valence: 89.7
Human Arousal: 39.6

GPT-4 Valence: 61.5
GPT-4 Arousal: 47

The image features a tiger resting under some dense foliage. It looks directly at the viewer with its distinct orange coat striped with black. The tiger appears formidable yet calm nestled among the greenery that provides a natural camouflage in its environment.

**Neutral**

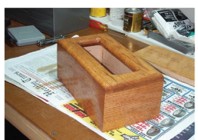

Human Valence: 54.5
Human Arousal: 18.4

GPT-4 Valence: 50
GPT-4 Arousal: 11

The image shows a white storage cabinet with multiple blue doors. Each door has a lock suggesting the cabinet is used for secure storage. One of the cabinet doors is open displaying an empty compartment inside. This type of cabinet is likely used in settings like offices or schools where secure storage is necessary.

Human Valence: 67.8
Human Arousal: 21.5

GPT-4 Valence: 50
GPT-4 Arousal: 16

The image shows a wooden object with a rectangular shape and a hollow center possibly a custom-made box or holder of some type sitting on a table. The table has various items on it including newspapers possibly some tools and other items typically found in a workshop setting. The wooden piece looks well-crafted with a nice finish indicating it might have either decorative or functional use such as a planter box or container.

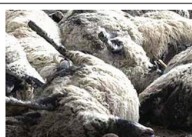

**Animal Mistreatment**

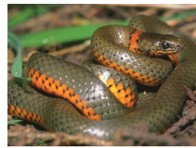

Human Valence: 26.3
Human Arousal: 54.7

GPT-4 Valence: 27
GPT-4 Arousal: 48.5

The image shows a dead antelope with long spiraled horns lying in a dry bushy landscape. It appears to be a hot arid environment typical of savannah or scrubland. The antelope seems to have been killed as evidenced by what looks like a wound on its side. The surrounding vegetation is sparse with small trees and shrubs scattered around the area.

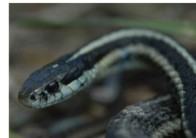

Human Valence: 7.1
Human Arousal: 77.5

GPT-4 Valence: 28
GPT-4 Arousal: 53.5

The image shows a white lion looking directly at the camera with its face partially framed by black bars suggesting that it is inside a cage. The lion is eyes appear calm, and its natural pale coloration is distinct with lighter fur compared to the typical golden color seen in most lions.

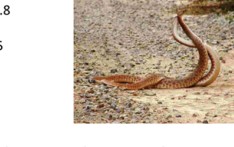

Human Valence: 11.7
Human Arousal: 56.4

GPT-4 Valence: 42.5
GPT-4 Arousal: 28.5

This image shows a group of sheep lying close together. Their wool appears dirty or muddy suggesting they might be in a rugged or outdoor environment. Some sheep have visible ear tags which are typically used for identification and tracking purposes in livestock management. The sheep look calm and are resting possibly huddling together for warmth or protection.

**Snakes**

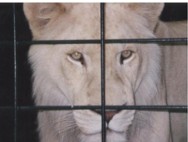

Human Valence: 40.1
Human Arousal: 55.8

GPT-4 Valence: 40
GPT-4 Arousal: 62

This image features a snake with a distinctive appearance. It has greenish-gray scales with a pattern of bright orange and light gray or white patches on its underbelly. It also has an orange band around its neck. The snake is coiled among some natural debris possibly leaves and twigs. This coloration and pattern suggest it may be a species that uses colorful markings as a warning or for camouflage in its natural habitat.

Human Valence: 52.4
Human Arousal: 43.8

GPT-4 Valence: 37.5
GPT-4 Arousal: 61

The image features a close-up of a garter snake. This snake is easily identifiable by its long slender body and the distinctive stripe patterns running along its length. it is likely in a natural habitat possibly among leaves or ground cover providing camouflage. The focus primarily on the head and upper body allows for a clear view of its features including the scales and eyes.

Human Valence: 63.7
Human Arousal: 45.3

GPT-4 Valence: 34.5
GPT-4 Arousal: 67

This image shows two snakes in what appears to be a combat pose likely engaged in a ritualistic dance or fight that is typical behavior during mating seasons or territorial disputes. The snakes are positioned upright with their bodies intertwined and heads raised suggesting a display of strength and dominance. The background features a dirt ground with small rocks and sparse vegetation which suggests a dry perhaps arid environment.

**Fig 2. Examples of images across different categories with textual descriptions.** Each category includes an example image demonstrating close agreement between human and GPT-4 ratings (difference ≤ 10), along with examples showing moderate agreement (difference between 10 and 20) and low agreement (difference >20). All images are sourced from [4].

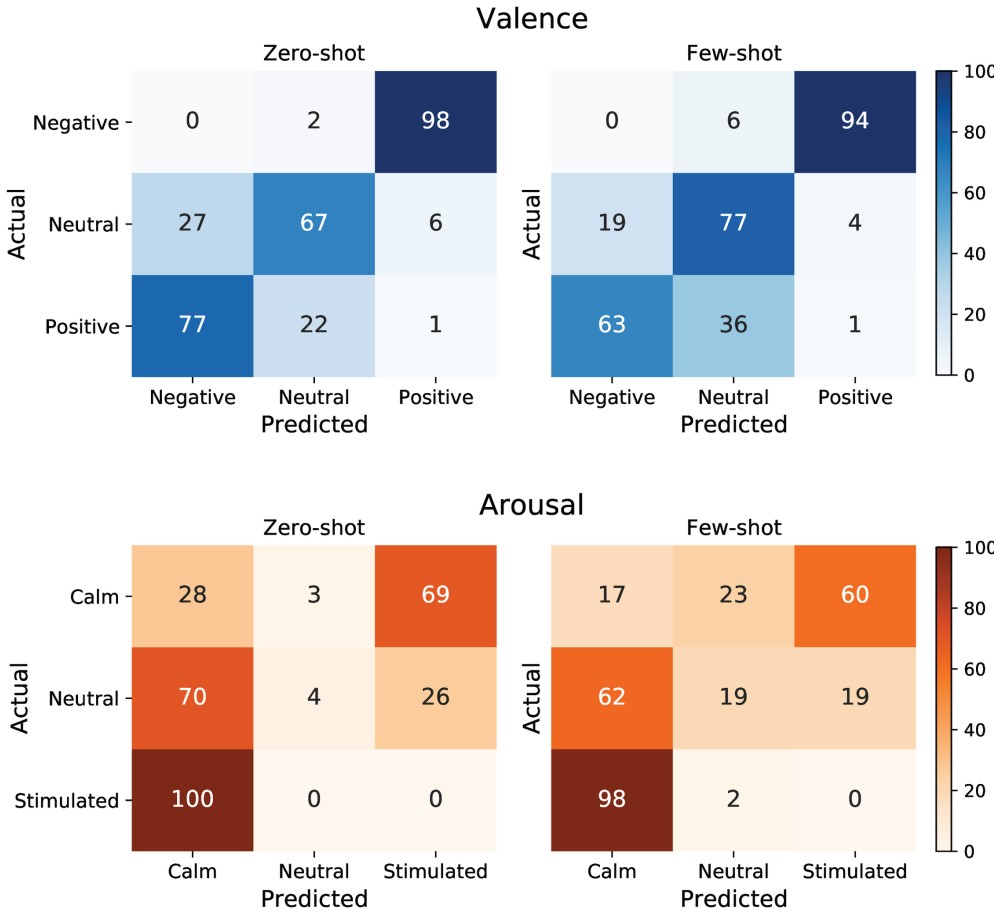

**Fig 3. Confusion matrix for 3-point Likert scale prompts for images.** Confusion matrix for 3-point Likert scale prompts under zero-shot and few-shot prompting. Values are expressed as percentages.

For valence, the top left matrix in Fig. 3 reveals a high accuracy in predicting Positive (98%) and Negative (77%) images. However, approximately 27% of Neutral valence ratings were misclassified as Negative and 6% as Positive. The inclusion of examples in few-shot prompting improved the model's ability to predict Neutral valence, increasing accuracy from 67% to 77%. However, this approach has led to a slight reduction in the prediction accuracy for Positive and Negative images.

Similarly, for arousal, the model performed well in predicting Calm (100%) and Stimulated (69%) states under zero-shot prompting, as shown in Fig. 3. Few-shot prompting further enhanced accuracy for Neutral arousal predictions but had no significant effect on predictions for Calm or Stimulated states.

S2 Fig and S3 Fig provide detailed confusion matrices, showing GPT-4's performance for each image category under both zero-shot and few-shot conditions. Tables 4 and 5 present the valence and arousal classification performance metrics for zero-shot and few-shot prompting across various image categories. Each metric evaluates the model's ability to classify emotional valence categories (Negative, Neutral, or Positive) or arousal states (Calm, Neutral, or Stimulated). Performance is assessed using both a two-class classification approach for individual categories and a three-class classification approach for all images combined. The

**Table 4. Valence classification performance metrics for zero-shot and few-shot prompting across different image categories.**

| Image category | Zero-shot | | | | Few-shot | | | |
|---|---|---|---|---|---|---|---|---|
| | Accuracy | Precision | Recall | F1-score | Accuracy | Precision | Recall | F1-score |
| All images | 0.77 | 0.78 | 0.80 | 0.79 | 0.73 | 0.77 | 0.78 | 0.77 |
| Positive | 0.98 | 1.0 | 0.98 | 0.99 | 0.94 | 1.0 | 0.94 | 0.97 |
| Neutral | 0.97 | 1.0 | 0.97 | 0.98 | 1.0 | 1.0 | 1.0 | 1.0 |
| Animal Mistreatment | 0.85 | 0.94 | 0.89 | 0.91 | 0.81 | 0.93 | 0.85 | 0.89 |
| Human Concerns | 0.81 | 0.95 | 0.77 | 0.85 | 0.78 | 0.98 | 0.70 | 0.82 |
| Snakes | 0.57 | 0.48 | 0.47 | 0.48 | 0.57 | 0.46 | 0.23 | 0.31 |
| Spiders | 0.65 | 0.73 | 0.79 | 0.76 | 0.52 | 0.71 | 0.55 | 0.62 |

**Table 5. Arousal classification performance metrics for zero-shot and few-shot prompting across different image categories.**

| Image category | Zero-shot | | | | Few-shot | | | |
|---|---|---|---|---|---|---|---|---|
| | Accuracy | Precision | Recall | F1-score | Accuracy | Precision | Recall | F1-score |
| All images | 0.57 | 0.51 | 0.58 | 0.44 | 0.55 | 0.50 | 0.59 | 0.49 |
| Positive | 0.64 | 0.63 | 1.00 | 0.77 | 0.62 | 0.62 | 0.97 | 0.76 |
| Neutral | 0.46 | 0.00 | 0.00 | 0.00 | 0.47 | 1.00 | 0.02 | 0.04 |
| Animal Mistreatment | 0.67 | 0.94 | 0.66 | 0.78 | 0.66 | 0.94 | 0.66 | 0.78 |
| Human Concerns | 0.81 | 0.95 | 0.77 | 0.85 | 0.78 | 0.98 | 0.70 | 0.82 |
| Snakes | 0.59 | 0.76 | 0.67 | 0.71 | 0.49 | 0.77 | 0.44 | 0.56 |
| Spiders | 0.76 | 0.85 | 0.87 | 0.86 | 0.65 | 0.84 | 0.72 | 0.78 |

two-class approach is used because, within individual categories (e.g., Negative), images are only ever rated as Negative or Neutral by both human raters and GPT-4, with Positive ratings not observed. The three-class approach is necessary for all images combined, as all three classes (Negative, Neutral, and Positive) are present.

For valence, the accuracy results in Table 4 show that GPT-4, without any prior training (zero-shot), correctly classified over 75% of the images in most categories, with the exception of Snakes and Spiders. Overall, zero-shot prompting consistently outperformed few-shot prompting across most metrics, including accuracy, recall, and F1-score. This indicates that the zero-shot approach was more reliable for classifying valence across a diverse set of images, providing a more consistent performance across different categories. Few-shot prompting, while still effective, showed slightly lower classification performance, suggesting that additional examples did not significantly improve the model's ability to generalize across the evaluated image categories.

Table 5 shows moderate accuracy for both zero-shot (0.57) and few-shot (0.55), indicating challenges in classifying arousal levels across all image categories. The high recall for Positive images for both zero-shot and few-shot indicate that most positive arousal instances were correctly identified. The zero-shot performance is very poor for the Neutral image category, indicating that the model failed to identify any Neutral cases correctly. Few-shot shows a slight improvement. Both zero-shot and few-shot prompting perform similarly for Negative image categories. Precision is notably high for both Animal Mistreatment and Human Concerns, but the recall is lower, suggesting that the model is more cautious in predicting high arousal but may miss some cases. The relatively higher F1-score in Snakes and Spiders suggest that GPT-4 is more effective in identifying high-arousal cases associated with those images.

Upon examining Negative images (Animal Mistreatment, Human Concerns, Snakes, and Spiders), human ratings fluctuated between Neutral and Negative, indicating that some images did not convey intensely negative emotions. According to the 3-point scale rating, 42% of these images received a Negative valence rating from humans, while the remaining 58%

were rated as Neutral. In comparison, GPT-4 classified 41% of the images as Negative, with the rest rated as Neutral and occasionally as Positive. Notably, the ratings do not always align between human raters and GPT-4. Images rated as Positive by GPT-4 within this category (3% of Negative images) were often misinterpreted. For example, GPT-4 failed to recognize that an image depicted an act of killing or that a group of individuals were refugees.

## Valence and arousal ratings for image descriptions

**Numeric response rating.** GPT-4's performance on numeric response rating tasks for valence and arousal, based on image descriptions, achieved a Pearson correlation of 0.79 ($p<0.001$) for valence and 0.65 ($p<0.001$) for arousal in the zero-shot setting. In the few-shot setting, the Pearson correlations were 0.78 ($p<0.001$) for valence and 0.71 ($p<0.001$) for arousal. The detailed results are presented in Tables 6 and 7, respectively.

Tables 6 and 7 demonstrate that GPT-4 closely approximates human ratings for both valence and arousal across zero-shot and few-shot settings. The MAE in the zero-shot setting shows that GPT-4 performs well in most categories. However, the inclusion of examples in the few-shot setting did not consistently improve valence ratings for negative image categories. For instance, the MAE for valence in Positive images decreases from 9.2 in the zero-shot setting to 6.5 in the few-shot setting, whereas in the Human Concerns category, it increases from 19.8 to 22.4. In contrast, for arousal, the few-shot setting generally led to a reduction in MAE across nearly all image categories. Fig 4 shows the distribution of differences in valence and arousal ratings between GPT-4 and human raters across all images and individual image categories, using kernel density estimates.

**Table 6. Numeric response ratings for valence across image descriptions under zero-shot and few-shot learning conditions. SD represents standard deviation; MAE represents mean absolute error.**

| Image category | Humans | | | GPT-4 zero-shot | | | | GPT-4 few-shot | | | |
|---|---|---|---|---|---|---|---|---|---|---|---|
| | Mean | SD | Range | Mean | SD | Range | MAE | Mean | SD | Range | MAE |
| All images | 44.5 | 25.2 | 0.4-98.7 | 47.4 | 20.4 | 10.0-95.0 | 11.8 | 50.5 | 20.9 | 5.0-96.1 | 12.4 |
| Positive | 89.6 | 6.2 | 71.9-98.7 | 81.1 | 6.9 | 61.0-95.0 | 9.2 | 85.1 | 7.3 | 64.8-96.1 | 6.5 |
| Neutral | 55.8 | 6.1 | 41.0-68.9 | 56.2 | 10.2 | 33.0-72.6 | 7.8 | 58.1 | 7.6 | 42.1-75.5 | 6.5 |
| Animal Mistreatment | 21.3 | 12.4 | 0.4-49.5 | 34.3 | 15.4 | 11.0-77.0 | 15.1 | 37.2 | 17.1 | 5.0-85.9 | 17.5 |
| Human Concerns | 28.0 | 17.5 | 0.7-61.4 | 45.9 | 21.7 | 10.0-86.0 | 19.8 | 49.3 | 21.5 | 10.9-87.8 | 22.4 |
| Snakes | 41.5 | 11.2 | 16.7-63.7 | 39.2 | 7.7 | 24.0-64.0 | 10.9 | 42.1 | 7.9 | 21.8-68.4 | 11.3 |
| Spiders | 35.1 | 11.4 | 9.5-57.0 | 35.0 | 5.1 | 22.0-61.0 | 9.1 | 37.8 | 6.6 | 15.3-58.4 | 10.4 |

**Table 7. Numeric response ratings for arousal across image descriptions under zero-shot and few-shot learning conditions. SD represents standard deviation; MAE represents mean absolute error.**

| Image category | Humans | | | GPT-4 zero-shot | | | | GPT-4 few-shot | | | |
|---|---|---|---|---|---|---|---|---|---|---|---|
| | Mean | SD | Range | Mean | SD | Range | MAE | Mean | SD | Range | MAE |
| All images | 47.7 | 19.5 | 5.9-92.4 | 48.3 | 13.6 | 10.0-82.0 | 12.1 | 46.0 | 14.6 | 13.5-83.9 | 11.2 |
| Positive | 21.6 | 10.7 | 5.9-66.0 | 37.4 | 8.1 | 25.0-74.0 | 16.3 | 31.1 | 9.6 | 13.5-75.1 | 11.7 |
| Neutral | 24.9 | 7.8 | 10.2-43.8 | 30.4 | 6.4 | 10.0-47.6 | 8.4 | 27.2 | 6.5 | 15.3-44.3 | 6.9 |
| Animal Mistreatment | 60.6 | 12.0 | 28.4-89.0 | 51.8 | 11.8 | 28.0-78.0 | 12.2 | 52.9 | 11.8 | 25.9, 83.4 | 11.9 |
| Human Concerns | 58.7 | 15.0 | 32.3-92.4 | 48.9 | 13.0 | 28.0-79.0 | 13.8 | 49.1 | 13.4 | 21.0-83.9 | 14.0 |
| Snakes | 53.6 | 10.7 | 30.6-72.9 | 57.2 | 8.9 | 33.0-82.0 | 11.2 | 53.7 | 9.0 | 33.3-80.7 | 11.4 |
| Spiders | 58.2 | 10.3 | 37.4-78.4 | 56.2 | 9.2 | 30.0-74.0 | 10.4 | 54.2 | 7.8 | 34.2-71.6 | 10.7 |

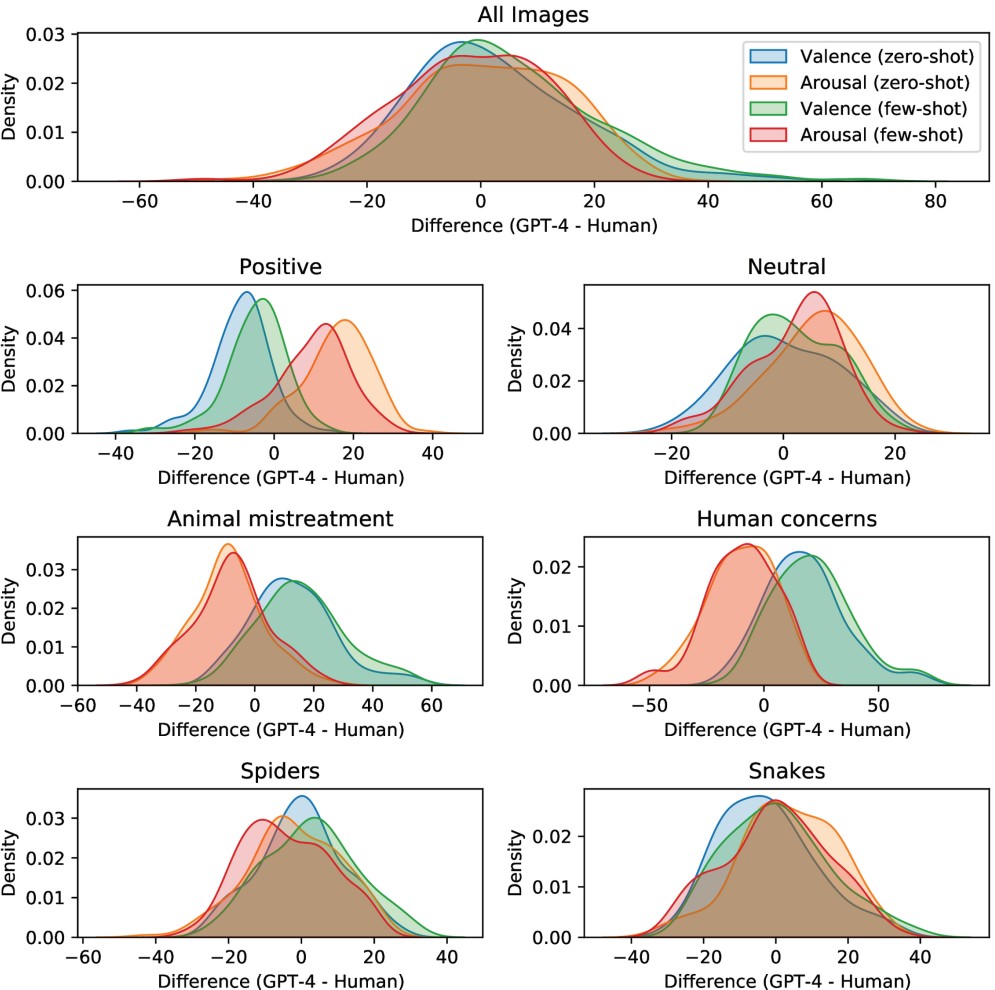

**Fig 4. Rating differences for image descriptions across image categories.** Distribution of differences in valence and arousal ratings between humans and GPT-4 for image descriptions across image categories.

**Likert scale rating.** Fig 5 shows the agreement between human and GPT-4 valence and arousal ratings of all images on the 3-point Likert scale, under zero-shot and few-shot conditions, respectively. Valence ratings are comparable under zero-shot and few-shot prompting. The model is well predicting the Positive and Neutral images.

Similar to the valence zero-shot matrix, Fig. 3 shows that GPT-4 performs well in correctly predicting Calm and Stimulated arousal states, achieving 100% and 69% accuracy, respectively, under zero-shot prompting. Providing examples improved the model's accuracy in predicting Neutral images, but had no significant effect on its performance for Calm or Stimulated arousal states.

Tables 8 and 9 present the valence and arousal classification performance metrics for zero-shot and few-shot prompting based on image descriptions. The results reveal patterns similar to those observed in the image rating task. For instance, the accuracy results in Table 8 show that GPT-4 correctly classified over 70% of the descriptions in most categories, with the exception of Snakes and Spiders. Overall, zero-shot and few-shot prompting exhibit comparable performance across most evaluation metrics. Additionally, Table 9 highlights

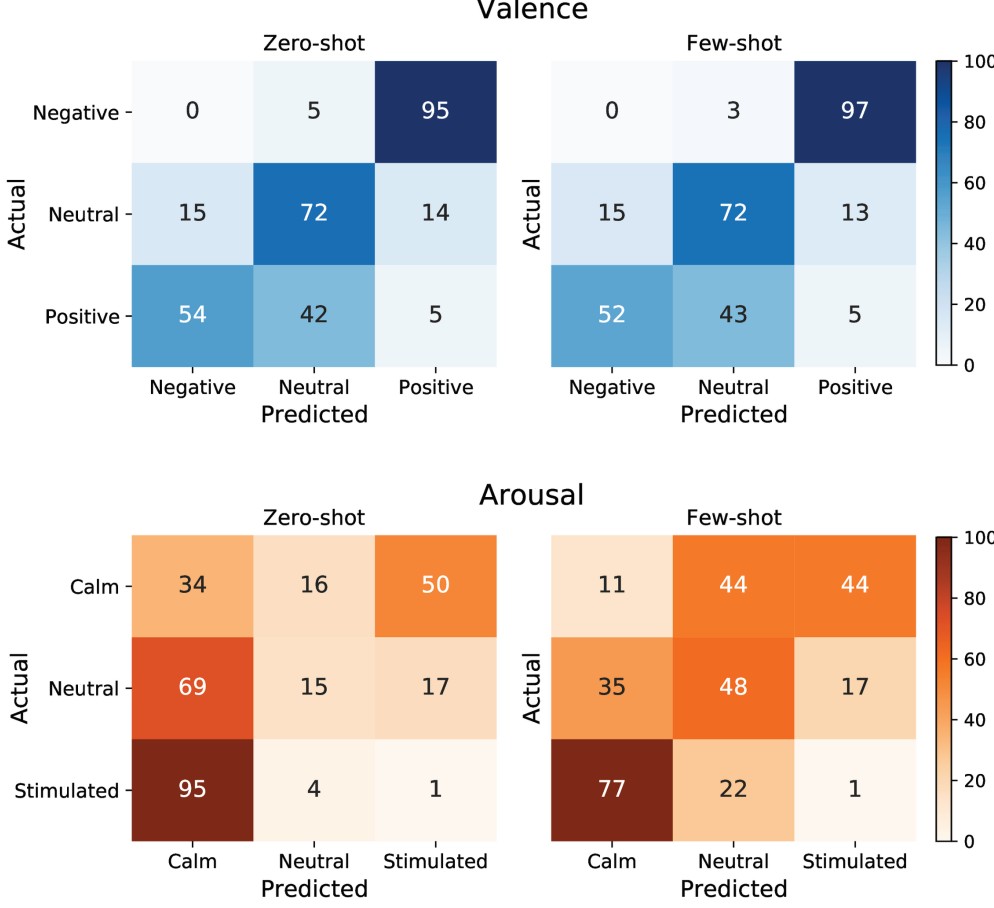

**Fig 5. Confusion matrix for 3-point Likert scale prompts for image descriptions.** Confusion matrix for 3-point Likert scale prompts under zero-shot and few-shot prompting. Values are expressed as percentages.

**Table 8. Valence classification performance metrics for zero-shot and few-shot prompting across different image categories.**

| Image category | Zero-shot | | | | Few-shot | | | |
|---|---|---|---|---|---|---|---|---|
| | Accuracy | Precision | Recall | F1-score | Accuracy | Precision | Recall | F1-score |
| All images | 0.67 | 0.69 | 0.74 | 0.69 | 0.66 | 0.69 | 0.73 | 0.67 |
| Positive | 0.95 | 1.00 | 0.95 | 0.98 | 0.97 | 1.00 | 0.97 | 0.98 |
| Neutral | 0.85 | 1.00 | 0.85 | 0.92 | 0.89 | 1.00 | 0.89 | 0.94 |
| Animal Mistreatment | 0.78 | 0.98 | 0.77 | 0.86 | 0.76 | 0.99 | 0.73 | 0.84 |
| Human Concerns | 0.68 | 0.94 | 0.58 | 0.72 | 0.68 | 0.98 | 0.56 | 0.71 |
| Snakes | 0.56 | 0.45 | 0.25 | 0.33 | 0.54 | 0.39 | 0.20 | 0.27 |
| Spiders | 0.49 | 0.76 | 0.41 | 0.53 | 0.47 | 0.71 | 0.42 | 0.53 |

moderate accuracy for both prompting conditions, suggesting persistent challenges in classifying arousal levels from image descriptions.

Figures S4 Fig and S5 Fig show the performance of GPT-4 against human ratings for each image description across different image categories under zero-shot and few-shot prompting.

**Table 9. Arousal classification performance metrics for zero-shot and few-shot prompting across different image categories.**

| Image category | Zero-shot | | | | Few-shot | | | |
|---|---|---|---|---|---|---|---|---|
| | Accuracy | Precision | Recall | F1-score | Accuracy | Precision | Recall | F1-score |
| All images | 0.48 | 0.48 | 0.53 | 0.42 | 0.50 | 0.53 | 0.56 | 0.50 |
| Positive | 0.67 | 0.66 | 0.92 | 0.77 | 0.65 | 0.69 | 0.76 | 0.72 |
| Neutral | 0.49 | 1.00 | 0.06 | 0.11 | 0.58 | 0.69 | 0.40 | 0.51 |
| Animal Mistreatment | 0.52 | 0.96 | 0.48 | 0.64 | 0.50 | 0.96 | 0.45 | 0.61 |
| Human Concerns | 0.54 | 0.95 | 0.43 | 0.59 | 0.51 | 0.97 | 0.38 | 0.55 |
| Snakes | 0.50 | 0.76 | 0.47 | 0.58 | 0.46 | 0.73 | 0.43 | 0.54 |
| Spiders | 0.59 | 0.91 | 0.57 | 0.70 | 0.49 | 0.83 | 0.47 | 0.60 |

## Discussion

The findings of this study highlight GPT-4's promising capacity to interpret emotional cues from images, particularly in assessing valence and arousal. The model's close alignment with human ratings demonstrates its utility in validating emotion-elicitation stimuli. These results suggest that GPT-4 could serve as a reliable tool for automating emotion-elicitation frameworks, potentially reducing the time and labor demands associated with traditional methods.

The study also highlighted some challenges in GPT-4's performance. Nonetheless, when GPT-4's ratings were less accurate, the discrepancies were typically minor, often involving shifts of just one point on the Likert scale. Notably, GPT-4 never misclassified an image from the Negative category as Positive, except in instances where it failed to capture critical image details.

A notable issue is the model's occasional inability to fully interpret the content of images, as evidenced by its textual descriptions. For example, GPT-4 misinterpreted a burn around a child's mouth as food residue and interpreted a man chained to a wooden pole as "pretending" for humor. In the Animal Mistreatment category, an image of dead sheep was described as "laying close together," failing to convey the emotional gravity of the scene (see Fig 2). These errors, though relatively infrequent (occurring in less than 2% of the images), highlight the challenges LLMs face in interpreting images.

Interestingly, GPT-4's performance varied significantly across different image categories. For instance, it excelled with images containing clear, positive emotional signals such as images of smiling babies or cute baby animals. In contrast, its accuracy decreased with images that required more nuanced emotional interpretations or where cultural perceptions might influence the emotional assessment, such as images of adult animals or complex human social scenes.

In other categories like Animal Mistreatment and Human Concerns, GPT-4 generally aligned well with human ratings when the scenes overtly depicted suffering or distress. However, it struggled with subtler contexts, such as images of refugees smiling, where the underlying emotional tones of hardship or resilience might not be immediately apparent. This indicates a need for LLMs to develop a deeper understanding of context and the less overt emotional undertones present in complex images.

In the Snakes and Spiders categories, GPT-4 never rated these images as Positive. It frequently rated images of these creatures in peaceful, natural settings (e.g., spiders weaving webs or non-venomous snakes coiled quietly) as Neutral. In contrast, human raters often reacted more negatively due to instinctive aversions. Conversely, GPT-4 reliably identified threats in dynamic or close-up scenes, such as spiders hunting prey or snakes poised to strike, rating them as Negative even when some human raters assigned Neutral ratings.

GPT-4's interpretation of the arousal scale diverges from typical human assessments. For images categorized as Neutral, GPT-4 frequently assigned an arousal rating of Calm, indicating a possible misconception of arousal as a measure of emotional intensity. This discrepancy underscores a fundamental misalignment between GPT-4's internal emotional framework and the evaluative scales commonly used by humans.

Moreover, while GPT-4 showed proficiency in deriving emotional insights from textual descriptions of images, it generally performed slightly better when analyzing the images directly. This may reflect the model's difficulty in capturing nuanced emotional contexts within textual descriptions, where it may over rely on the sentiment of individual words rather than the overall emotional context.

One key observation was that including examples in few-shot learning scenarios did not consistently improve GPT-4's performance, particularly with negative images. This inconsistency can be attributed to the wide variability in human ratings. For instance, in the Snakes category, 59% of the images were rated as Neutral and 41% as Negative based on their valence.

## Study limitations

It is crucial to acknowledge that the numerical scales for valence and arousal used in this study are not standard in everyday discussions of human emotions. This deviation might have impacted the outcomes, considering GPT-4's training largely comprises datasets where such specific formats are infrequent.

Our analysis was primarily concentrated on GPT-4 due to its notable performance. Nonetheless, to obtain a comprehensive understanding, it is vital to include a diverse range of models, compare their performances, and delineate their respective strengths and weaknesses.

Additionally, while GPT-4 has shown impressive capabilities in various image emotion rating tasks, its closed-source nature poses limitations on the transparency and interpretability of its outputs. The inaccessibility to the underlying model architecture and training data restricts our ability to fully comprehend the factors influencing its judgments, which is essential for thorough analysis and validation.

Finally, detecting hallucinations in emotion rating tasks can be complex. One potential method to enhance reliability is prompting the model to justify each rating. Analyzing these justifications in relation to their corresponding ratings could uncover inconsistencies or unsupported responses. Future research could explore this approach to enhance the robustness of emotion rating tasks.

## Conclusion

The study successfully demonstrates GPT-4's potential in automating the process of emotion elicitation using visual stimuli, particularly focusing on the dimensions of valence and arousal. By comparing GPT-4's performance against human evaluations, we found that the model approximates human ratings with high accuracy under both zero-shot and few-shot conditions. This suggests that GPT-4 can serve as a valuable tool in streamlining the selection and validation of emotional stimuli, potentially reducing the labor and time required in traditional methods.

However, the study also reveals some challenges in GPT-4's ability to interpret more nuanced emotional cues in images, particularly those requiring a deeper understanding of context or subtle emotional undertones. This is evident from occasional discrepancies in the model's ratings, especially in images depicting complex human emotions or social scenes.

Despite these challenges, GPT-4's robust performance highlights its utility in not only enhancing current emotional analysis frameworks but also in expanding the possibilities

for their application in real-world scenarios. More effort should aim to refine these models further, ensuring that they can handle a wider variety of emotional contexts with greater sensitivity and accuracy.

Future research should explore the emotional elicitation across additional dimensions, test against a broader range of image datasets, and incorporate more large language models into the analysis. While this study deliberately avoided detailed prompting to preserve an unbiased assessment of GPT-4's inherent capabilities, subsequent studies could investigate how contextual prompts may improve alignment with human emotional evaluations. Additionally, integrating multimodal data and enriching model training with diverse emotional datasets could yield more nuanced insights.

## Supporting information

**S1 Fig. Prompt templates for few-shot prompting.** Top: few-shot prompt for images. Bottom: few-shot prompt for image descriptions.
(TIFF)

**S2 Fig. Confusion matrix for 3-point Likert scale ratings across image categories under zero-shot prompting.** Top: valence ratings. Bottom: arousal ratings.
(TIFF)

**S3 Fig. Confusion matrix for 3-point Likert scale ratings across image categories under few-shot prompting.** Top: valence ratings. Bottom: arousal ratings.
(TIFF)

**S4 Fig. Confusion matrix for 3-point Likert scale ratings across image description categories under zero-shot prompting.** Top: valence ratings. Bottom: arousal ratings.
(TIFF)

**S5 Fig. Confusion matrix for 3-point Likert scale ratings across image description categories under few-shot prompting.** Top: valence ratings. Bottom: arousal ratings.
(TIFF)

## Author contributions

**Conceptualization:** Hend Alrasheed, Sharifa Alghowinem.

**Formal analysis:** Hend Alrasheed, Adwa Alghihab, Sharifa Alghowinem.

**Investigation:** Adwa Alghihab, Sharifa Alghowinem.

**Methodology:** Hend Alrasheed.

**Project administration:** Hend Alrasheed, Alex Pentland.

**Resources:** Hend Alrasheed, Alex Pentland, Sharifa Alghowinem.

**Supervision:** Alex Pentland.

**Validation:** Hend Alrasheed, Adwa Alghihab, Alex Pentland.

**Visualization:** Hend Alrasheed.

**Writing – original draft:** Hend Alrasheed, Sharifa Alghowinem.

**Writing – review & editing:** Hend Alrasheed, Sharifa Alghowinem.

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
