## [Decision Letter · Decision Letter 0]

PONE-D-24-58179Evaluating the capacity of large language models to interpret emotions in imagesPLOS ONE

Dear Dr. Alrasheed,

Thank you for submitting your manuscript to <em data-end="199" data-start="189">PLOS ONE</em>. We appreciate the effort and thought that went into this research, and we are pleased to inform you that we would like to invite you to submit a revised version of your paper. Based on the reviewers’ assessments, we are offering a **revise and resubmit with minor revisions**.

Both reviewers acknowledge the relevance and significance of your study, particularly in evaluating GPT-4’s capabilities in recognizing and rating emotions from visual stimuli. Below is a summary of their key comments that I encourage you to address in your revision:

<h3 data-end="781" data-start="753">**Reviewer Comments:**</h3>

**Importance of Research Scope:** Reviewer 1 highlighted that your study aligns with applied research trends in emotion recognition and its potential broad applicability. Reviewer 2 suggested strengthening the introduction by explicitly explaining why evaluating emotional cues from **non-facial images** is important.**Clarity in Numeric Scale Tables:** Reviewer 2 noted that the numeric scale tables could be difficult to interpret because they assume prior knowledge about differences between LLM and human ratings. They suggest including a comparison metric to indicate what constitutes a “close” comparison to human ratings and providing a brief explanation of its significance.**Contextual Emotion Recognition Comparison:** The manuscript focuses more on facial emotion recognition, but Reviewer 2 suggests discussing **contextual emotion recognition** and referencing other studies that have used the same dataset, even if they do not involve LLMs.**Figure Placement & Labeling:** The placement of figures at the end of the paper makes it difficult for readers to understand their context. Reviewer 2 recommends integrating figures into the relevant sections where they are mentioned in the text and improving figure labels to clarify their significance.**Demographic Details of Human Raters:** Given that perceived emotional effects can vary across demographics, Reviewer 2 recommends providing more details on the demographics of human raters to address potential biases in the study.

<h3 data-end="2350" data-start="2311">**Additional Editorial Request:**</h3>

We also noticed that the **image quality in the manuscript is not optimal**, as they appear blurred in the current submission. Please ensure that all figures are of **high resolution** and check that they remain clear after uploading the PDF version.

Overall, your study makes an important contribution to the field, and we are confident that these refinements will enhance its clarity and impact. We look forward to receiving your revised manuscript and thank you for your efforts in addressing these minor revisions.

Congratulations on your work, and we appreciate your contribution to <em data-end="2955" data-start="2945">PLOS ONE</em>.

We look forward to receiving your revised manuscript.

Kind regards,

Carlos Carrasco-Farré

Academic Editor

PLOS ONE

Journal Requirements:

2.  Please ensure that you refer to Figure 4 in your text as, if accepted, production will need this reference to link the reader to the figure.

3. We notice that your supplementary figures are uploaded with the file type 'Figure'. Please amend the file type to 'Supporting Information'. Please ensure that each Supporting Information file has a legend listed in the manuscript after the references list.

4. We note you have included a table to which you do not refer in the text of your manuscript. Please ensure that you refer to Table 8 and 9 in your text; if accepted, production will need this reference to link the reader to the Table.

Reviewers' comments:

Reviewer's Responses to Questions

**Comments to the Author**

1. Is the manuscript technically sound, and do the data support the conclusions?

Reviewer #1: Yes

Reviewer #2: Yes

2. Has the statistical analysis been performed appropriately and rigorously? 

Reviewer #1: Yes

Reviewer #2: Yes

3. Have the authors made all data underlying the findings in their manuscript fully available?

Reviewer #1: Yes

Reviewer #2: Yes

4. Is the manuscript presented in an intelligible fashion and written in standard English?

Reviewer #1: Yes

Reviewer #2: Yes

5. Review Comments to the Author

Reviewer #1: Evaluating GPT-4's capability to recognize and rate emotions from visual stimuli is a critical area of research, as it has the potential to become one of the preferred methods in applied emotional and psychological studies. This evaluation is important for several reasons:

+ GPT-4's ability to interpret visual stimuli, such as facial expressions, can provide deeper insights into emotional dynamics across various contexts, from individual well-being to social interactions.

+ The use of GPT-4 for emotion recognition aligns with trends in applied research methodologies, offering scalable, efficient, and adaptable tools for large-scale studies in fields like marketing, sociology, education, and behavioral science.

+ Rigorous testing and evaluation ensure the reliability of GPT-4 in academic and applied research, encouraging broader adoption in scientific communities.

By focusing on these aspects, the evaluation of GPT-4's emotion recognition capabilities could establish it as a versatile and trusted tool in understanding human emotions through visual stimuli.

Reviewer #2: Summary:

* This research is attempting to evaluate the effectiveness of Chat-GPT4o in predicting the emotional response different images have on the subject through both visual and written depiction of images in order to determine if Chat-GPT is as good or better than previous work on explicitly engineered emotional evaluation of direct subject observance using deep learning and deterministic models.

Strengths:

* The paper did a good job of providing multiple evaluation criteria for their central thesis

* I was happy to see all the visual representations of the data and its comparisons within the paper

* I felt the paper made clear it’s conclusion and did a good job supporting that conclusion by highlighting the relevant data

Weaknesses:

* The paper did not establish why evaluation of emotional queues from non-facial images was important.

* The tables on the numeric scale are difficult to evaluate because it assumes the reader understands what difference between the LLM and the human raters is low versus high. 

* The image based related work seems to focus more on facial emotion recognition and not contextual emotion recognition. A well known and highly available dataset is used in this work for this purpose, however it is not compared with existing work. Are there any relevant work related to the same dataset being used? Even if they are not LLMs?

* Other than the tables, each figure is pasted at the end of the paper and the labels of the figures do not provide appropriate context to show what each represents.

Suggestions for strengthening the paper:

* Include a paragraph of why this research matters and how it can benefit others or be expanded to practical applications.

* Include a comparison metric in the numeric scale tables that indicate what is a “close” comparison to the human rating and then briefly explain how the comparison metric is used in the paper.

* Include at least one mention of a study done with the same dataset and evaluation criteria that was used in your research (i.e. likert scale emotion detection)

* It would help the reader better understand the context of each figure in the paper if the figure was shown in the same area as it was mentioned in the paper. Pasting them all at the end makes it very difficult to determine what each figure is intended to show as the labeling does not provide that information. The reader must go into the paper and find where each figure is mentioned.

* There is no mention of what demographic the human raters are. Since this deals with perceived emotional effects of specific images, having a skewed demographic can influence the ratings. Thus, it would be effective to have a diverse demographic participating.

6. PLOS authors have the option to publish the peer review history of their article (what does this mean?). If published, this will include your full peer review and any attached files.

Reviewer #1: No

Reviewer #2: No

---

## [Author Response · Author response to Decision Letter 1]

4 Apr 2025

Dear Dr. Carrasco-Farré,

Thank you for the opportunity to submit a revised version of our manuscript, Evaluating the Capacity of Large Language Models to Interpret Emotions in Images, to PLOS ONE. We sincerely appreciate the time and effort that you and the reviewers have dedicated to providing thoughtful and constructive feedback.

We have carefully addressed the reviewers' major concerns and incorporated their suggestions throughout the manuscript to improve clarity and readability. In addition, we have addressed the journal’s formatting and submission requirements as outlined. Specifically:

• We enhanced the quality of the figures to ensure greater clarity.

• We added in-text references to all figures and tables.

• We changed the file type of the supplementary figures to comply with submission guidelines.

• We reviewed and updated the reference list by removing two duplicate entries and several references that were not cited in the current version of the manuscript. Additional references were included in response to the reviewers’ suggestions. We also revised the formatting of all entries to align with PLOS ONE style.

We have also addressed all reviewer comments and suggestions. A detailed, point-by-point response is provided below. The corresponding revisions are clearly marked in red in the tracked version of the manuscript.

Thank you again for considering our resubmission. We look forward to your response.

Sincerely,

Hend Alrasheed

Media Lab, Massachusetts Institute of Technology

hrasheed@mit.edu

April 4, 2025

Point-by-point response to the reviewers’ comments and concerns.

Response to Reviewer 1 Comments

We thank the reviewer for their reading of the manuscript and for their constructive comments.

Comment 1.1: Evaluating GPT-4's capability to recognize and rate emotions from visual stimuli is a critical area of research, as it has the potential to become one of the preferred methods in applied emotional and psychological studies. This evaluation is important for several reasons:

- GPT-4's ability to interpret visual stimuli, such as facial expressions, can provide deeper insights into emotional dynamics across various contexts, from individual well-being to social interactions.

- The use of GPT-4 for emotion recognition aligns with trends in applied research methodologies, offering scalable, efficient, and adaptable tools for large-scale studies in fields like marketing, sociology, education, and behavioural science.

- Rigorous testing and evaluation ensure the reliability of GPT-4 in academic and applied research, encouraging broader adoption in scientific communities.

By focusing on these aspects, the evaluation of GPT-4's emotion recognition capabilities could establish it as a versatile and trusted tool in understanding human emotions through visual stimuli.

Response 1.1: We appreciate the reviewer’s thoughtful comments highlighting the significance of evaluating GPT-4’s ability to recognize and rate emotions from visual stimuli.

Response to Reviewer 2 Comments

We thank the reviewer for their reading of the manuscript and for their constructive comments. We have taken the comments into consideration to improve the quality of the manuscript. Please find below a point by point response to each comment.

Comment 2.1: The paper did not establish why evaluation of emotional queues from non-facial images was important.

Response 2.1: We have added a paragraph to the Introduction that clarifies the importance of evaluating emotional cues from non-facial images (lines: 149-161).

“While most efforts in the literature focus on evaluating the capabilities of LLMs in extracting emotions from facial images, our work focus to emotion recognition from general, non-facial images, such as objects, environments, animals, and abstract scenes. Despite their rich emotional content and widespread use in psychological, affective computing, and mental health research \cite{Geneva,ortis2020survey}, the interpretation of emotions elicited by non-facial imagery remains relatively underexplored, particularly in the context of Large Language Models. Assessing emotional responses to such stimuli is crucial, as they provide opportunities to study affective processing in broader, more ecologically valid contexts where facial expressions may be absent or irrelevant. Moreover, non-facial images are a foundational component of standardized emotional elicitation datasets such as GAPED and IAPS, highlighting their significance in emotion research.”

Comment 2.2: The tables on the numeric scale are difficult to evaluate because it assumes the reader understands what difference between the LLM and the human raters is low versus high.

Include a comparison metric in the numeric scale tables that indicate what is a “close” comparison to the human rating and then briefly explain how the comparison metric is used in the paper.

Response 2.2: We have added a paragraph to the Results section clarifying how the difference between GPT-4 and human ratings should be interpreted using the Mean Absolute Error (MAE) metric. The new text explains the MAE values relative to the 0–100 rating scale (lines: 356-360).

“Note that the maximum possible deviation between GPT-4 and the human rating for any image is 100, as both valence and arousal were rated on a 0–100 scale. In our results, MAE values ranged from approximately 5 to 15, indicating that the average error across all image categories was relatively small, representing only 5–15\% of the maximum possible error.”

Comment 2.3: The image based related work seems to focus more on facial emotion recognition and not contextual emotion recognition. A well known and highly available dataset is used in this work for this purpose, however it is not compared with existing work. Are there any relevant work related to the same dataset being used? Even if they are not LLMs?

Response 2.3: We have added three studies that utilize the GAPED image dataset in distinct emotion elicitation tasks to the Related Work section (lines: 151-158).

“While most efforts in the literature focus on evaluating the capabilities of LLMs in extracting emotions from facial images, our work shifts the focus to emotion recognition from general, non-facial images. Accordingly, we use the GAPED image dataset, a well-established resource for emotion elicitation. Several studies have employed this dataset for a range of emotional research tasks. For example, Moyal et al. \cite{moyal2018Categorized} used GAPED images to elicit discrete emotions such as fear, disgust, sadness, and happiness. Balsamo et al. \cite{Balsamo2020bottom} used valence and arousal ratings to evaluate the effectiveness of GAPED images in evoking emotional responses. Moreover, the author in \cite{Brainerd2018emotional} used GAPED images to explore how different combinations of valence and arousal influence cognitive processing, particularly in emotionally ambiguous contexts.”

Comment 2.4: Other than the tables, each figure is pasted at the end of the paper and the labels of the figures do not provide appropriate context to show what each represents. It would help the reader better understand the context of each figure in the paper if the figure was shown in the same area as it was mentioned in the paper. Pasting them all at the end makes it very difficult to determine what each figure is intended to show as the labeling does not provide that information. The reader must go into the paper and find where each figure is mentioned.

Response 2.4: We apologize for the inconvenience. In accordance with PLOS submission requirements, all figures have been uploaded as separate files and not embedded within the manuscript. We understand that figure links and placements will be handled during the final production stage.

Comment 2.5: Include a paragraph of why this research matters and how it can benefit others or be expanded to practical applications.

Response 2.5: We have added a paragraph to the end of the Introduction that highlights the importance of our work and its potential applications. (lines: 82-86).

“By demonstrating that GPT-4 can closely approximate human emotional ratings of visual stimuli, this work offers a scalable and efficient alternative to traditional emotion validation methods, which are often labor-intensive and costly. Such automation can streamline experimental design in psychology and facilitate the creation of emotionally intelligent AI agents.”

Comment 2.6: Include at least one mention of a study done with the same dataset and evaluation criteria that was used in your research (i.e. likert scale emotion detection).

Response 2.6: We have added a reference to a study that uses the same dataset to elicit valence and arousal ratings using a Likert scale, and have incorporated it to the Related Work section (lines: 158-161).

“A recent study \cite{Berezina2024} used the GAPED dataset to assess valence and arousal ratings within a Malaysian population using a 9-point Likert scale, with the goal of identifying culturally specific patterns in emotional responses.”

Comment 2.7: There is no mention of what demographic the human raters are. Since this deals with perceived emotional effects of specific images, having a skewed demographic can influence the ratings. Thus, it would be effective to have a diverse demographic participating.

Response 2.7: Thank you for this important observation. The demographic information of the human raters is provided in the manuscript in the Image Dataset section, based on the original GAPED dataset documentation. While we rely on GAPED’s existing ratings, we agree that future work should explore expanding demographic diversity to further enhance generalizability.

---

## [Editor Report · Decision Letter 1]

Evaluating the capacity of large language models to interpret emotions in images

PONE-D-24-58179R1

Dear Dr. Alrasheed,

We’re pleased to inform you that your manuscript has been judged scientifically suitable for publication and will be formally accepted for publication once it meets all outstanding technical requirements.

Kind regards,

Carlos Carrasco-Farré

Academic Editor

PLOS ONE

---

## [Editor Report · Acceptance letter]

PONE-D-24-58179R1

PLOS ONE

Dear Dr. Alrasheed,

I'm pleased to inform you that your manuscript has been deemed suitable for publication in PLOS ONE. Congratulations! Your manuscript is now being handed over to our production team.

Kind regards,

on behalf of

Dr. Carlos Carrasco-Farré

Academic Editor

PLOS ONE